# Spores of *Trichoderma* Strains over *P. vulgaris* Beans: Direct Effect on Insect Attacks and Indirect Effect on Agronomic Parameters

**DOI:** 10.3390/insects13121086

**Published:** 2022-11-25

**Authors:** Álvaro Rodríguez-González, Guzmán Carro-Huerga, Marcos Guerra, Sara Mayo-Prieto, Alejandra Juana Porteous-Álvarez, Alicia Lorenzana, María Piedad Campelo, Alexia Fernández-Marcos, Pedro Antonio Casquero, Santiago Gutiérrez

**Affiliations:** 1Grupo Universitario de Investigación en Ingeniería y Agricultura Sostenible (GUIIAS), Instituto de Medio, Ambiente Recursos Naturales y Biodiversidad (INMARENBIO), Escuela de Ingeniería Agraria y Forestal (EIAF), Universidad de León, 24071 León, Spain; 2Grupo Universitario de Investigación en Ingeniería y Agricultura Sostenible (GUIIAS), Escuela de Ingeniería Agraria y Forestal (EIAF), Campus de Ponferrada, Universidad de León, 24401 Ponferrada, Spain

**Keywords:** *Acanthoscelides obtectus*, *Phaseolus vulgaris*, *Trichoderma*, agronomic parameters

## Abstract

**Simple Summary:**

*Acanthoscelides obtectus* is an insect pest that attacks wild and cultivated common beans (*Phaseolus vulgaris* L). Four *Trichoderma* strains, *T. arundinaceum* IBT 40837, Ta37-17.139 (=Δ*tri*17), and Ta37-23.74 (=Δ*tri*23), and *T. brevicompactum* IBT 40841 were assessed to establish their direct effect on insect attacks and their indirect effect on the plants grown from the beans treated with those fungal strains and exposed to insect attacks. Treatments of bean seeds with different *Trichoderma* strains led to different survival rates in the insects. Insect cadavers (in contact with Δ*tri*23) showed growth of this strain. The emergence of insects was reduced in the beans treated with the Ta37, Tb41, and Δ*tri*17 strains. The undamaged beans (treated with Ta37 and Δ*tri*23) provided a high capacity of germination, whereas the Δ*tri*17 and Tb41 treatments increased the capacity of germination in the damaged beans. The undamaged beans treated with Δ*tri*23 obtained the greatest dry weights of the aerial part and root system in the plants. More studies on the mechanisms of insect control, plant growth promotion, and volatile compound production by Δ*tri*23 and Tb41 should be explored in order to commercialize these fungal species on a massive scale.

**Abstract:**

*Acanthoscelides obtectus* is an insect pest that attacks wild and cultivated common beans (*Phaseolus vulgaris* L). Four *Trichoderma* strains, the *T. arundinaceum* IBT 40837 wild-type strain (=Ta37), a producer of trichothecene harzianum A (HA), two transformants of *T. arundinaceum* strain, Ta37-17.139 (=Δ*tri*17) and Ta37-23.74 (=Δ*tri*23), and the *T. brevicompactum* IBT 40841 wild-type strain (=Tb41), which produces the trichothecene trichodermin, were assessed to establish their direct effect on insect attacks and their indirect effect on the plants grown from the beans treated with those fungal strains and exposed to insect attacks. Treatments of bean seeds with different *Trichoderma* strains led to different survival rates in the insects, and the Tb41 strain caused the lowest survival rate of all. An 86.10% of the insect cadavers (in contact with Δ*tri*23) showed growth of this strain. This was the treatment that attracted the greatest number of insects. The daily emergence was reduced in beans treated with the Ta37, Tb41, and Δ*tri*17 strains. The undamaged beans treated with Ta37 and Δ*tri*23 showed a high capacity of germination (80.00% and 75.00%, respectively), whereas the Δ*tri*17 and Tb41 treatments increased the capacity of germination in the damaged beans (66.67%). The undamaged beans treated with Δ*tri*23 had the greatest dry weights for the aerial part (4.22 g) and root system in the plants (0.62 g). More studies on the mechanisms of insect control, plant growth promotion, and trichodermol and trichodermin production by Δ*tri*23 and Tb41, respectively, should be explored in order to commercialize these fungal species on a large scale.

## 1. Introduction

*Beauveria bassiana* (Hypocreales, Cordycipitaceae) and *Metarhizium anisopliae* (Hypocreales, Clavicipitaceae) are two fungi capable of managing postharvest insects [1]. The most significant insect pests that cause damage to stored products are *Sithophilus zeamais* [2], *A. obtectus* [3], and *Callosobruchus maculatus* (F.) [4], which can be controlled with *B. bassiana*; and *Rhyzopertha dominica* [5] and *Sitophilus oryzae* [6], which can be controlled with *M. anisopliae*.

*Trichoderma* (Hypocreales, Hypocreaceae) is a well-characterized fungal genus that currently comprises more than 200 molecularly defined species [7]. *Trichoderma* species are cosmopolitan and prevalent components of different ecosystems in a wide range of climatic zones [8]. Several *Trichoderma* spp. are non-pathogenic soil-borne fungi that are recognized as opportunistic, avirulent, and plant symbionts capable of colonizing the root system of many plants [9]. This genus is known to provide significant advantages in agriculture for its capacity to protect crops against diseases and to generally improve crop yield [10]. Biocontrol using *Trichoderma* strains can be performed by mycoparasitism, antibiosis, or competition, but the most effective biocontrol strains use more than one of these strategies [11].

*Trichoderma* strains have an enormous potential to synthesize an extensive variety of secondary metabolites [12,13], such as pyrones [6-pentyl-2H-pyran-2-one (6-PP) derivatives] [14,15], butenolides [16], peptaibols [17,18], terpenes (e.g., trichothecenes, triterpenes) [19,20,21,22], and gliotoxin, viridin, harzianopyridone, and harziandione [13,23].

Terpenes include a plethora of compounds with a great variety of roles in mediating antagonistic and beneficial interactions between organisms [24]. However, there is a certain lack of information regarding the effect of terpene mycotoxins, e.g., volatile mycotoxins or intermediates, produced by saprophytic-beneficial fungi, on their interaction with plants and the herbivores that eat them [25].

Trichothecenes are a group of non-volatile sesquiterpene mycotoxins with a central core of fused cyclohexene/tetrahydropyran rings [26,27]. The majority of these compounds are phytotoxic, have significant antibiotic activity, and are highly toxic for humans and animals, irritating the skin or the intestinal mucous membrane, and affecting the immune and nervous systems [28]. Information about the plant–trichothecene interactions remains largely unknown, and it is thought that despite their phytotoxic activity, they also suppress the defense response in host plants [29]. Nevertheless, assays in plants showed that trichothecene harzianum A (HA), produced by *T. arundinaceum*, not only lacks phytotoxic activity but also induces the expression of the plant-defense-related genes belonging to the salicylic acid (SA) and jasmonic acid (JA) signaling pathways [20].

Bean weevil, *Acanthoscelides obtectus* (Coleoptera: Chrysomelidae: Bruchidae), is an insect pest that threatens wild and cultivated common beans (*Phaseolus vulgaris* L.) [30,31,32], both in the field and in storage, where the pest causes the highest number of losses [33]. Thus, if the initial population of *A. obtectus* insects is not controlled, it can grow exponentially and spoil the entire stored product [34]. Traditionally, several strategies such as the application of synthetic insecticides (phosphine, pyrethroids, and organophosphates) [35], or the use of physical barriers, e.g., hermetic packaging [36], have been used to prevent this pest from causing damage to stored beans.

However, issues about pest resistance and risk to human health or the environment have arisen in relation to the continued application of synthetic insecticides [37]. There is, therefore, a growing demand to look for more sustainable alternatives regarding pest control, including the use of essential oils [38,39], volatile compounds that intervene in the natural defense of the plant against insects [40,41], or the implementation of biological control agents such as filamentous fungi or bacteria [42,43,44,45,46,47]. Over the last few years, the utilization of fungi to control diverse pests and plant diseases has markedly increased, resulting in a great number of commercial products available [42,48], all of which are able to minimize the side effects not only on the indigenous-beneficial organisms but also on the environment.

The main objectives of this study were to determine how *A. obtectus* insect adults are affected by the terpene compounds produced by different wild-type strains of *T. arundinaceum* and *T. brevicompactum,* and two transformant strains of *T. arundinaceum*; how terpene production affects the *A. obtectus* insect adults; and also, the effect of these strains on the germination capacity and agronomic parameters of the plants obtained either from healthy beans or from beans previously damaged by *A. obtectus* larvae.

## 2. Materials and Methods

### 2.1. Fungal Strains

Two wild *Trichoderma* strains, *Trichoderma arundinaceum* IBT 40837 (=Ta37) (IBT = strains collected from the “Institut for Biotecknologi”), which produces trichothecene harzianum A (HA), *T. brevicompactum* IBT 40841 (=Tb41), producer of trichodermin [49], and two transformants of Ta37, Ta37-17.139 (Δ*tri*17) and Ta37-23.74 (Δ*tri*23), in which the genes tri17 and tri23 (related to the harzianum A biosynthetic pathway) were deleted, [22,49] were used in this experiment. The Δ*tri*17 and Δ*tri*23 mutants do not produce HA, but they accumulate trichodermol, an intermediate in the synthesis of HA [22,50]. The fungal strains were kept in the mycological collection of the “Grupo de Investigación en Ingeniería y Agricultura Sostenible (GUIIAS)”, located in the “Plant Pests and Diseases Laboratory”, Escuela de Ingeniería Agraria y Forestal (EIAF)—Universidad de Leon (ULE). Spores of all the fungal strains were conserved in a 50% glycerol suspension at −80 °C for long-term storage.

The *Trichoderma* strains were grown on a PPG solid medium (2% mashed potatoes (Nestlé, Vevey, Switzerland), 2% glucose (Panreac Applichem, Chicago IL, USA), 2% agar (Oxoid, Ltd., Hampshire, UK)) (Sigma-Aldrich Chemie GmbH, Steinheim, Germany)) and incubated in controlled conditions (described in Section 2.2.) for 7–10 days. To collect the spores, 20 mL of autoclaved sterile water was poured on the grown *Trichoderma* strains, and the surface was scrubbed with a brush. It was then placed under a microscope (200× magnification and a suspension of 1 × 10^7^ spores/mL), using a Neubauer chamber. The spore suspension was then put in Eppendorf tubes (1.5 mL) (EppendorfAG, Hamburg, Germany), which were stored at 20 ± 1 °C for a few hours until they were used in the experiments [43,47].

### 2.2. Insect Collection and Rearing

The original population of *A. obtectus* was collected from 2017 to 2019 in storage rooms inside the Protected Geographical Indication (PGI) “Alubia de La Bañeza-León”. The common bean (*P. vulgaris* L.) “Canela” variety was used to feed the different *A. obtectus* stages, which were put into glass jars (150 mm diameter, 250 mm high) and covered with a cloth to allow air to enter. Every 3 days, *A. obtectus* adults were removed from the glass jar with the beans in order to maintain a group of young adults (1–3 days old) to use in the experiments. Insects, before and after the treatments, were kept in a chamber with controlled temperature (24 ± 1 °C), humidity (60 ± 5%), and placed under a photoperiod of 16 h of light (luminous intensity of 1000 lux) and 8 h of darkness [43,47].

### 2.3. Design of Experiments

#### 2.3.1. Effects of Beans Sprayed with Spores of the *Trichoderma* Strains on Insects

One milliliter of the spore suspension (1 × 10^7^ spores/mL) of each *Trichoderma* strain was directly applied to 40 *P. vulgaris* beans placed in Petri dishes (90 mm diameter) (Sigma-Aldrich, Steinheim, Germany). Eight treatments (four *Trichoderma* strains and their respective four controls) with four replicates for each treatment were applied. Sterile water was used as the control treatment and as the carrier (support in which the spores of the fungi were diluted) in all the treatments with *Trichoderma* strains. The application of the treatments, *Trichoderma* strains, and their respective controls was carried out with a manual loading Potter Tower (Burkard Scientific Limited, P.O. Box 55 Uxbridge, Middx UB8 2RT, UK), according to the methodology of [51]. Once the treatments were applied to the beans, the beans were subsequently transferred to a structure made of 5 circular plastic containers according to the methodology described by Mazzonetto and Vendramim [52] and Fouad et al. [53]. There were four containers (A, B, C, and D) (40 mm diameter, 70 mm high) with a central container (E) (120 mm diameter, 60 mm high) connected to the other 4 containers by plastic cylinders (70 mm long, 7 mm diameter). In the structure, containers B and D were arranged diagonally and filled with 40 beans treated with the same *Trichoderma* strains, and containers A and C (also arranged diagonally) were filled with 40 beans treated with the respective control of the *Trichoderma* strain, which had previously been filled in containers B and D. In total, 20 insects (10 males and 10 females, 1 to 3 days old) were released into the central container (E). After 24 h, once the insects decided their location in containers A, B, C, or D, the beans (treated with *Trichoderma* strains or their controls) and insects were transferred back to the Petri dishes, where the treatments had been applied using the Potter Tower, as previously described. The mortality of the insects in contact with the beans (the insects were considered dead if there was no reaction when prodded) was recorded daily over a 30-day period. During the entire process of Experiment 1, the Petri dishes were kept in a chamber with the same controlled conditions as described in Section 2.2. Once all the dead insects were collected for each of the treatments, the insect cadavers were placed in Petri dishes with a Rose-Bengal Chloramphenicol Agar medium (4 insect cadavers/dish), to verify the presence of fungal strains in their bodies.

#### 2.3.2. Effect of the Fungal Strains on the Biological Development of the Insects

The insects that emerged from the beans in the Petri dishes (sprayed in Experiment 1) were used to estimate the effect of *Trichoderma* strains on the biological development of insects. The progeny of the insects that emerged from the beans (2nd generation) and the damaged beans (by the insect emergence holes) in each Petri dish were recorded daily over an 18-day period (day 1 of this period being the equivalent to day 31 after the treatments were applied) according to the methodology described by [43]. The insects that emerged were removed from the Petri dishes, and the daily insect emergence of the 2nd generation was recorded. The number of treatments, replicates, and controlled conditions were similar to those described previously in Section 2.3.1.

#### 2.3.3. Determination of Germination Capacity of Beans Attacked by Insects

This study was performed in a climatic chamber to test the germination capacity of the beans previously sprayed with *Trichoderma* strains or with sterile water (the control) and that were damaged or not damaged after being in contact with the insects, as was described in Section 2.3.1. Polypropylene pots (1-L capacity) with peat (TYPical, Brill, Georgsdorf, Germany) were used to make this bioassay. Each pot was filled with 250 mL of water prior to sowing. Five undamaged beans (one bean without any holes from each Petri dish, as described in Section 2.3.1.) were randomly selected and sown in five pots (one bean/pot). Five damaged beans (beans with at least one hole per bean caused by insect emergence) were selected and sown in five pots (one bean/pot). The culture was maintained for 45 days under controlled conditions with a photoperiod of 16/8 h (light/darkness), a temperature of 25 °C/16 °C (day/night), 60 ± 5% RH, and brightness of 3500 lux. They were watered every 4 days with 250 mL of tap water per pot according to the methodology described by Mayo et al. [54,55]. A nutrient solution was added during the 2nd–4th week according to the methodology described by [56]. Plant emergence was evaluated on the 11th and 17th day after sowing. Plants were then removed after 45 days, and the agronomic parameter “dry weight” (72 h in an oven, at 82 °C) of both the aerial part and the root system was evaluated.

### 2.4. Statistical Analysis

All the data analyzed were normally distributed and presented homoscedasticity and were subjected (in each of the experiments) to the statistical analyses described below.

The survival data of the insects were submitted to the Kaplan–Meier estimator, and the functions obtained from each treatment were compared using the log-rank test (Mantel-Cox) (*p* < 0.05).

A randomly completed experiment using the generalized linear model (GLM) procedure, with four treatments and four replicates was subjected to an ANOVA (IBM SPSS Statistics, Version 26.0.). Differences (*p* < 0.05) in all the evaluated parameters were examined by mean comparisons using the Fisher least significant difference (LSD) tests.

A randomly completed experiment following a generalized linear model (GLM) procedure, with eight treatments and twenty replicates (undamaged or damaged beans) was subjected to an ANOVA (IBM SPSS Statistics, Version 26.0.). Differences (*p* < 0.05) in all the evaluated parameters were examined by mean comparisons using the Fisher least significant difference (LSD) test.

## 3. Results

### 3.1. Effect of Beans Sprayed with Spores of Trichoderma Strains on the Survival Rate of Insects

The type of treatment influenced the survival rate of the insects (log-rank χ^2^ = 25.019; df = 7141; *p* < 0.001).

The number and survival rate of the insects exposed to the beans sprayed with different *Trichoderma* strains were 15 insects with a mean of 18.00 ± 1.00 days for the Ta37 strain, 22 insects with a mean of 21.00 ± 3.00 days for the Δ*tri*23 strain, 16 insects with a mean of 19.50 ± 1.50 days for the Δ*tri*17 strain, and 14 insects with a mean of 21.00 ± 3.00 days for the Tb41 strain.

The number and survival rate of the insects exposed to the beans sprayed with sterile water (controls) were 20 insects with a mean of 19.00 ± 1.00 days for Ta37 (control), 17 insects with a mean of 20.80 ± 0.37 days for Δ*tri*23 (control), 23 insects with a mean of 19.00 ± 0.45 days for Δ*tri*17 (control), and 22 insects with a mean of 18.75 ± 0.31 days for Tb41 (control).

Only the 14 insects subjected to the Tb41 strain had a significantly (F = 8.000; df = 1.34; *p* < 0.005) lower life expectancy (17.00 days) than those obtained with Tb41 (control), 18.75 days. In addition, the Tb41 strain and Tb41 (control) treatments (log-rank χ^2^ = 25.019; df = 7141; *p* < 0.001) lowered life expectancy significantly more than those obtained with the Δ*tri*23 strain and Δ*tri*23 (control), 21.00 and 20.80 days, respectively.

### 3.2. Effect of Trichoderma Strains on the Biological Development of Insects

#### 3.2.1. Daily Emergence Curves of Insects Exposed to Different Trichoderma Strains

The beans treated with Ta37 and Δ*tri17* saw a reduction in insect emergence on the 1st day of the evaluation (means of 0.25 and 0.50 insects, respectively) compared with those obtained for their respective controls on the same day (Figure 1a,c). Similarly, the Tb41 treatment provided a significant reduction in insect emergence in comparison to its control, but its effect lasted longer than the rest of the treatments, so differences were significant on days 1, 2, 9, 15, and 16 (Figure 1d).

By contrast, insect emergence from the beans treated with Δ*tri23* on days 1, 2, 3, 8, and 13 was significantly greater in comparison to the Δ*tri23* control (Figure 1b).

#### 3.2.2. Fungal Growth on Insect Cadavers

The insect cadavers showed the growth of *Trichoderma* strains when placed on the Rose-Bengal Chloramphenicol Agar medium after being in contact with the beans sprayed with *Trichoderma* strains and their controls.

The number of insect cadavers and the growth percentage of the *Trichoderma* strains on them were 15 cadavers with a mean of 64.60 ± 22.10% for the Ta37 strain (Figure 2a), 22 cadavers with a mean of 86.10 ± 8.30% for the Δ*tri*23 strain (Figure 2b), 16 cadavers with a mean of 37.50 ± 14.20% for the Δ*tri*17 strain (Figure 2c), and 14 cadavers with a mean of 17.50 ± 11.80% for the Tb41 strain (Figure 2d).

The number of insect cadavers and the growth percentage of the *Trichoderma* strains on them were 20 cadavers with a mean of 0.00 ± 0.00% in the Ta37 (control), 17 cadavers with a mean of 0.00 ± 0.00% for the Δ*tri*23 (control), 23 cadavers with a mean of 0.00 ± 0.00% for the Δ*tri*17 (control), and 22 cadavers with a mean of 0.00 ± 0.00% for the Tb41 (control).

The percentages of the *Trichoderma* strains grown in the cadavers that were previously in contact with the beans treated with fungal isolates were significantly higher than those obtained for their respective controls (F = 8.504; df = 1.6; *p* = 0.027 in the Ta37 strain; F = 106.780; df = 1.6; *p* ≤ 0.001 in the Δ*tri*23 strain; F = 6.943; df = 1.6; *p* = 0.039 in the Δ*tri*17 strain; and F = 4.194; df = 1.6; *p* = 0.048 in the Tb41 strain). Only the percentages of *Trichoderma* growth in the cadavers that were previously in contact with the beans treated with the Δ*tri*23 strain were significantly higher (F = 4.022; df = 3.12; *p* = 0.034) than those obtained for the other strains.

### 3.3. Effect of Insects on the Germination of Beans

The germination rates (% ± SE) of the *P. vulgaris*-undamaged beans sprayed with *Trichoderma* strains were 80.00 ± 9.17% for the Ta37 strain, 75.00 ± 9.93% for the Δ*tri*23 strain, 70.00 ± 10.51% for the Δ*tri*17 strain, and 45.00 ± 11.41% for the Tb41 strain. The germination rates of the damaged beans were with no data for the Ta37 strain, 62.50 ± 18.29% for the Δ*tri*23 strain, 66.67 ± 33.33% for the Δ*tri*17 strain, and 66.67 ± 33.33% for the Tb41 strain.

The germination rates (% ± SE) of the *P. vulgaris*-undamaged beans sprayed with the controls were 70.00 ± 10.51% for Ta37 (control), 65.00 ± 10.94% for Δ*tri*23 (control), 60.00 ± 11.23% for Δ*tri*17 (control), and 65.00 ± 10.94% for Tb41 (control). The germination rates of the damaged beans were 0.00 ± 0.00% with no data for Ta37 (control), 100.00 ± 0.00% for Δ*tri*23 (control), 87.50 ± 12.50% for Δ*tri*17 (control), and 55.56 ± 17.57% for Tb41 (control).

The undamaged beans sprayed with Ta37 and Δ*tri*23 reached higher germination rates (80.00 ± 9.17% and 75.00 ± 9.93%, respectively) than the beans treated with Tb41 (45.00 ± 11.41%). No significant differences were found between the undamaged beans sprayed with the fungal strains and their respective controls in relation to their germination capacity.

The damaged beans sprayed with Ta37 (control) reached the lowest germination rate of all the treatments, significantly lower than the germination rates for the rest of the control treatments, Δ*tri23* (control) and Δ*tri17* (control). Significant differences between the damaged beans sprayed with the fungal strains and their respective controls were only found for the beans sprayed with the Δ*tri23* strain, whose germination rate was significantly (F = 3.246; df = 3.18; *p* = 0.046) lower (62.50 ± 18.29%) than that for Δ*tri23* (control).

### 3.4. Other Parameters Evaluated in the Plants after 45 Days

#### 3.4.1. The Effect of Fungal Strains on the Agronomic Parameters of the Plants Grown from Treated Beans

The undamaged beans sprayed with the Δ*tri23* spores had the greatest dry weights for the aerial part and root system of their plants (4.22 g and 0.62 g, respectively), significantly higher than those from the plants grown from the beans treated with the rest of the fungal strains (Ta37, Δ*tri17,* and Tb41). The plants grown from the undamaged Δ*tri23*-sprayed beans had a greater weight for the aerial part than its control (Δ*tri23* control), whereas the undamaged Ta37- and Δ*tri17*-sprayed beans had weights for the aerial part of the plant significantly lower than those obtained for their control treatments. Only the undamaged beans sprayed with Δ*tri23* resulted in plants with a dry weight for the root system that was significantly higher than those achieved for its control treatment (Δ*tri23* control) (Table 1).

Regarding the damaged beans, no significant differences were found for any of the agronomic parameters evaluated, either among treatments or between each of the fungal strains and their respective controls (Table 1).

#### 3.4.2. The Effect of Seed Condition on the Agronomic Parameters of the Plants Obtained from the Treated Beans

The plants grown from the undamaged beans sprayed with Δ*tri23* spores had significantly greater dry weights for the aerial part and the root system (4.22 and 0.62 g, respectively) than those obtained from the plants whose beans were sprayed with the same strain but had previously been damaged by insects (Figure 3a). Additionally, the undamaged beans sprayed with Tb41 produced plants with a significantly greater dry weight for the root system (0.49 g) than the plants obtained from the beans sprayed with the same strain but had previously been damaged by the larvae (Figure 3b).

With regard to the beans sprayed with sterile water (control), the results indicated that the undamaged beans sprayed with the Δ*tri17*-control treatment produced plants with significantly greater dry weights for the aerial part and the root system (3.91 and 0.52 g, respectively), than those obtained from the plants whose beans were sprayed with the Δ*tri17*-control treatment but had previously been damaged by the larvae (Figure 3c). In addition, the undamaged beans sprayed with sterile water (Δ*tri23* control) produced plants with a significantly heavier dry weight for the root system (0.53 g), than the plants obtained from the beans sprayed with the Δ*tri23*-control treatment but that had previously been damaged by the larvae (Figure 3d).

## 4. Discussion

The application of *Trichoderma* strains on beans affected the survival rate of the insects in contact with those seeds. The Tb41 strain significantly reduced the insects’ survival rate to 17 days, whereas the insects in contact with the Δ*tri23* strain had the greatest survival rate (more than 21 days). This longer insect life span implies that they had a longer reproduction period, which resulted in a greater emergence of insects in the next generation, so the insect emergence from the beans treated with Δ*tri23* was significantly greater (in 5 out of the 18 days in which it was evaluated) than that reached by its respective control. Other reports showed a modification of insect development because their hosts (seeds) were treated with fungi. Thus, Akello and Siroka [57] reported that the inoculation of bean seeds with fungal isolates (one of them being *T. asperellum* M2RT4) reduced the population of *Acyrthosiphon pisum* (Homoptera: Aphididae) by 33-fold, compared with the population growth observed in the untreated seeds. Menjivar Barahona [58] found that there was a reduction in the whitefly population in tomato seeds inoculated with *T. atroviride*. Most of the biopesticides and biofertilizers currently available on the market are based on the beneficial symbionts of the *Trichoderma* genus [59]. Several reports have also proven the potential of *Trichoderma* spp. as a natural control agent against some targeted insects, such as *Spodoptera littoralis* (Lepidoptera: Noctuidae), *Tenebrio molitor* (Coleoptera: Tenebrionidae) and *Xylosandrus crassiusculus* (Coleoptera: Curculionidae) [60,61,62]. More recently, Rodríguez-González et al. [42,47] described the reduction in insect survival with *T. citrinoviride* and *T. harzianum,* and *T. brevicompactum* (Tb41), one of the wild-type strains evaluated in their work, which is based on the results of this work. Therefore, it might be suitable for the control of this insect pest, due to the reduction in the survival rate of adult insects and the consequent reduction in the number of insects that emerge from the beans in the next generation.

The insect cadavers showed external growth of the *Trichoderma* strains used for the bean inoculation. The insect cadavers that were previously in contact with the Δ*tri*23 strain showed the greatest percentage of fungal growth (86.10%). This strain, which accumulates trichodermol, was the one that attracted the greatest number of insects (22 adults). The movement of the insects to the beans treated with the spores of *Trichoderma* strains or to the volatiles produced by them has recently been discovered [43,45,47]. Furthermore, the insects in contact with the untreated beans (controls) did not show any external growth of the fungal strains when their cadavers were evaluated. The fungal growth observed in the cadavers of *A. obtectus* in the previous contact with the treated beans would support the idea that the *Trichoderma* was the cause of death. These fungi have unique invasion mechanisms that allow them to penetrate the cuticle or the wall of the digestive tract of the insects directly, which makes them excellent biological control agents that act as contact insecticides [63]. *Trichoderma*, unlike other agents, does not need to be ingested by the insect, but its infection occurs through the contact and adhesion of the spores to the buccal parts, intersegmental membranes, or through the spiracles of the insects [64]. The fungal spores germinate in the host’s cuticle, penetrate, and spread throughout the body. After the fungus kills the insect, it can grow and sporulate out of the corpse, increasing the likelihood that other insects may be infected [65]. There are numerous studies on the use of entomopathogenic microorganisms, such as fungi, which have great potential as controlling agents because of their ability to cause disease and insect death [66].

The capacity of the germination of the beans varied depending on the *Trichoderma* strains that were applied and on the seed condition. Thus, the undamaged beans treated with Ta37, Δ*tri23,* and Δ*tri17* exhibited a greater capacity of germination than their respective controls, while the damaged Tb41-treated beans had a greater capacity of germination than their control at the same stages. In both cases, Δ*tri17* significantly protected the beans, which was achieved by lowering the survival rate of the insects in contact with the Δ*tri17*-treated beans, and subsequently favored the germination of the damaged beans, as long as the seed’s embryo was not damaged by the *A. obtectus* larvae.

The use of biological control agents, e.g., *T. harzianum* strains, is one of the alternatives for seed treatment while aiming for greater sustainability in agriculture [67,68]. A wide variety of *T. harzianum* strains have been used in seed treatment but little is known about the possible interactions between *Trichoderma* spp. and the early stages of seed germination [69] as well as the dosage needed. To avoid the effect of antagonists on seed germination and vigor, Dalzotto et al. [68] obtained a rate of germination 4% higher in the *P. vulgaris* seeds treated with *Trichoderma*, than the percentage of the germination observed in the control treatment. Mastouri et al. [69] showed that the inoculation of tomato seeds with *T. harzianum* T-22 (Rifai) conidia favored seed germination and growth in in vitro cultures. The increase in the germination of the seeds treated with *Trichoderma* strains has also been attributed to the production of phytohormones from *Trichoderma*, which would be responsible for that effect [68]. According to López-Bucio et al. [70], *T. harzianum* produces harzianic acid and isoharzianic acid, which promote plant growth. On the other hand, the excessive production of indole acetic acid (IAA), ethylene [71], auxins, and cytokinin hormones [72] inhibit plant cell division and elongation, impairing germination and the development of seedlings.

Finally, the application of *Trichoderma* strains on the bean seeds had a diverse effect on the dry weight of the plants grown from seed after 45 days. Thus, the application of the Δ*tri*23 strain on the undamaged beans resulted in plants with significantly higher dry weights for the aerial part and the root system than their respective controls and to the other *Trichoderma* strains that were analyzed in this work. Moreover, the weights of the plants grown from the beans previously damaged by *A. obtectus* larvae and treated with the Δ*tri*23 strain were significantly higher in comparison to the rest of the treatments. Previous studies have also described the effect of the application of *Trichoderma* on the improvement in seed germination, vegetative growth, and flowering in horticultural crops such as cucumber, periwinkle, chrysanthemum, and lettuce [73,74,75]. The application of the *Trichoderma* inoculum at an early stage of crop growth maximizes the benefits in terms of root development and nutrient uptake. Furthermore, increases in plant growth following *Trichoderma* treatment depend on the specific crop or plant genotype [70]. Inoculation with *Trichoderma* spp. in crops such as cucumber, maize, bean, and tomato [76,77,78,79,80,81,82] showed increases in root growth and shoot biomass production (increases in dry weight, shoot length, and leaf area), whereas *T. harzianum* (T-969 isolate) and *Trichoderma* spp. applied directly to tomato seeds produced plants with a great shoot height, shoot diameter, and fresh and dry shoot weights [81,82].

## 5. Conclusions

Δ*tri23* growth was observed in 86.10% of the insect cadavers that had previously been in contact with this strain, and it was the treatment that attracted the greatest number of insects. The daily emergence of the insects was reduced in the beans treated with Ta37, Δ*tri17*, and Tb41. The undamaged beans treated with Ta37 or Δ*tri*23 provided a great germination capacity, whereas the Δ*tri17* and Tb41 treatments increased the germination capacity of damaged beans. The undamaged beans treated with Δ*tri23* had the greatest dry weights for the aerial part and root system in the plants. More studies on the mechanisms of *A. obtectus* control, *P. vulgaris* plant growth promotion, and trichodermol and trichodermin production by Δ*tri23* and Tb41 strains should be explored for these fungal strains to be commercialized on a massive scale without causing any harmful effects on human health and environment.

## Figures and Tables

**Figure 1 insects-13-01086-f001:**
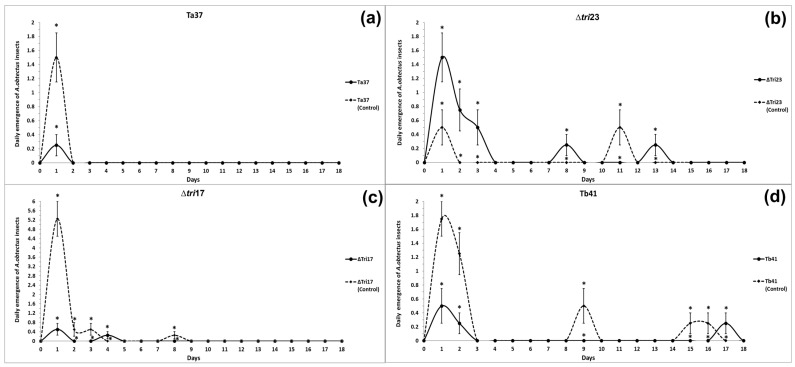
Daily emergence of *A. obtectus* adults from beans treated with different *Trichoderma* strains: (**a**) Ta37, (**b**) Δ*tri23*, (**c**) Δ*tri17,* and (**d**) Tb41. The values represent the mean of four replicates for each fungal isolate. The vertical bars represent the standard error (SE). Asterisks indicate significant differences for a specific date.

**Figure 2 insects-13-01086-f002:**
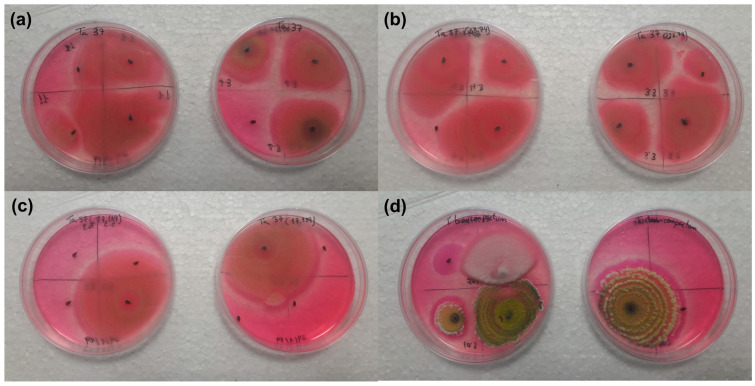
Growth of different *Trichoderma* strains on insect cadavers placed on the Rose-Bengal Chloramphenicol Agar medium: (**a**) Ta37, (**b**) Δ*tri23*, (**c**) Δ*tri17*, and (**d**) Tb41.

**Figure 3 insects-13-01086-f003:**
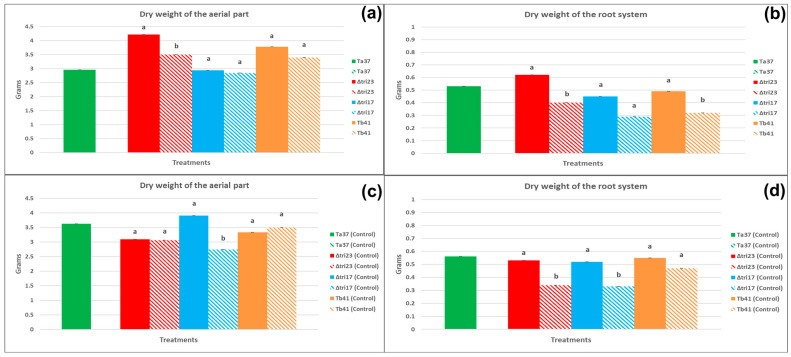
The effect of seed condition on the agronomic parameters (mean in grams ± SE) of the plants (45 days) grown from beans sprayed with different *Trichoderma* strains and their controls. Bars with pure color represent undamaged beans and bars with faded color represent damaged beans: (**a**) the dry weight of the aerial part of the plants grown from beans treated with *Trichoderma* strains; (**b**) the dry weight of the root system of the plants grown from beans treated with *Trichoderma* strains; (**c**) the dry weight of the aerial part of plants grown from beans treated with sterile water; (**d**) the dry weight of the root system of plants grown from beans treated with sterile water. Different lowercase letters indicate significant differences between a fungal isolate, in pure color, and its control, in faded color (=beans sprayed with sterile water), within each treatment and parameter evaluated; LSD test at 0.05.

**Table 1 insects-13-01086-t001:** The effect of fungal strains on the agronomic parameters (mean in grams ± SE) of the plants (45 days) grown from the beans sprayed with *Trichoderma* strains and their controls.

Treatments		Undamaged Beans					Damaged Beans			
		Dry Weight of the Arial Part (g)		Dry Weight of the Root System (g)			Dry Weight of the Aerial Part (g)		Dry Weight of the Root System (g)	
Ta37		2.96 ± 0.21bB ^a,b^		0.53 ± 0.05aAB ^a,b^			-		-	
Ta37 (Control)		3.63 ± 0.23aAB		0.56 ± 0.05aA			-		-	
Δ*tri23*		4.22 ± 0.23aA		0.62 ± 0.05aA			3.50 ± 0.21aA ^a,b^		0.40 ± 0.06aA ^a,b^	
Δ*tri23* (Control)		3.10 ± 0.26bB		0.53 ± 0.05bA			3.06 ± 0.21aA		0.34 ± 0.02aA	
Δ*tri17*		2.94 ± 0.26bB		0.45 ± 0.04aB			2.85 ± 0.15aA		0.29 ± 0.05aA	
Δ*tri17* (Control)		3.91 ± 0.35aA		0.52 ± 0.05aA			2.74 ± 0.33aA		0.33 ± 0.07aA	
Tb41		3.53 ± 0.29aB		0.48 ± 0.04aAB			3.40 ± 0.10aA		0.32 ± 0.04aA	
Tb41 (Control)		3.44 ± 0.15aAB		0.57 ± 0.04aA			3.50 ± 0.18aA		0.47 ± 0.03aA	
		*Trichoderma*	Control	*Trichoderma*	Control		*Trichoderma*	Control	*Trichoderma*	Control
	F	6.661	2.868	2.928	0.134	F	1.952	2.062	0.716	1.740
	df	(3.52)	(3.53)	(3.52)	(3.53)	df	(2.6)	(2.13)	(2.6)	(2.13)
	*p*	<0.001	0.046	0.042	0.939	*p*	0.222	0.167	0.526	0.214

^a^ Different lowercase letters indicate significant differences between *Trichoderma* strains and their respective controls within each parameter and seed condition; LSD test at 0.05. ^b^ Different capital letters indicate significant differences among *Trichoderma* strains (on one hand) or among controls (=beans sprayed with sterile water) (on the other hand) within each parameter and seed condition; LSD test at 0.05.

## Data Availability

The data presented in this study are available in this manuscript.

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
