# Peer review of "Spores of Trichoderma Strains over P. vulgaris Beans: Direct Effect on Insect Attacks and Indirect Effect on Agronomic Parameters"

_insects, 2022, doi:10.3390/insects13121086_

Round 1

Reviewer 1 Report

The Ms by Rodríguez-González, reports on the effects of Trichoderma strains on the bean weevil Acanthoscelides obtectus.

The Ms suffers from various serious problem. First of all, it is replicating a study from the same group that has been published in 2018 (see https://doi.org/10.1016/j.jspr.2018.05.001) with no novelty involved in the present study. Besides the idea replication, the Ms format is almost identical repeating the same section and some of the paragraphs (especially in the introduction and methods) are almost identical.

Secondly, the method section is not well described and many details are lacking. Furthermore, the authors divided their experiments in 3 bioassays, while in fact all the three are linked and the second and third experiments are a kind of follow up of the first using material ( insect and seed) resulting from that first bioassay...this raises serious doubt on the independence of the bioassays and potential biases ( also in statistical analysis)!!   

Moreover, the scientific writing is poor and in many instance does not allow understanding neither how things were done nor the rationale behind what was done. For examples, many scientific names are not italicized (see the minors below), some parts of the methodology were not well referenced (lines 156-159, see the minors below)!!  In the result section ( 3.2.2. and table 2), the authors present some findings of bioassay that was not described in the methodology!!!

Based on the above, I do not recommend publishing the Ms in its present form.

Minors:

line 40: replace "provided" with "showed"

line 45: replace "massive" with "large"

line 47: put the scientific names in italic

line 54: change the word "mechanisms"

lines 50-65: As the focus of the paper is on the Trichoderma,  I suggest to move this section to the end of the introduction..just before the paragraph stating the objectives of the study. Doing so, the introduction should be rearranged ( exp: lines 58-71)

lines 68-71: rephrase these sentences.

line 83: correct to "synthesize"

line 112: put the scientific name in italic.

line 127: put the scientific name in italic.

Lines 136-144: I suggest to joint the 2.3 section with the 2.1 section.

line 141: correct the concentration of spores.

line 149: correct the concentration of spores.

line 150: put the scientific name in italic

lines 156-159: the methodology here described was first used by other authors (Mazzonetto and Vendramim 2003 and Fouad et al.,2012) different from the references cited here!!! The ethics rules clearly recommend to cite the original reference!!

Lines 166-169: were the petri dishes also treated with the same strain of trichoderma??

line169: delete the word "insect"

Author Response

Manuscript ID: Insects - 1952235

Title: Spores of wild and transformed Trichoderma strains over P. vulgaris beans: direct effect on Acanthoscelides obtectus attacks and indirect effect on agronomic parameters

Open Review

English language and style

( ) Extensive editing of English language and style required
(x) Moderate English changes required
( ) English language and style are fine/minor spell check required
( ) I don't feel qualified to judge about the English language and style

In the last version of the attached manuscript the English has been checked by a native English speaker so the text has been improved.

Yes

Can be improved

Must be improved

Not applicable

Does the introduction provide sufficient background and include all relevant references?

( )

( )

(x)

( )

Are all the cited references relevant to the research?

( )

( )

(x)

( )

Is the research design appropriate?

( )

( )

(x)

( )

Are the methods adequately described?

( )

( )

(x)

( )

Are the results clearly presented?

( )

( )

(x)

( )

Are the conclusions supported by the results?

( )

( )

(x)

( )

Report to Reviewer 1:

Comments and Suggestions for Authors

The Ms by Rodríguez-González, reports on the effects of Trichoderma strains on the bean weevil Acanthoscelides obtectus.

The Ms suffers from various serious problem. First of all, it is replicating a study from the same group that has been published in 2018 (see https://doi.org/10.1016/j.jspr.2018.05.001) with no novelty involved in the present study. Besides the idea replication, the Ms format is almost identical repeating the same section and some of the paragraphs (especially in the introduction and methods) are almost identical.

Dear Editor, we appreciate very much your decision, because it gives us the opportunity to highlight the differences between the recently submitted study and the paper previously published by our group (doi: 10.1016/j.jspr.2018.05.001) in the year 2018.

Several works were done in the past to describe the effect of the different trichothecene biosynthetic intermediates in plants. Trichothecene analogs are well characterized as mycotoxins exhibiting important phytotoxic activities, but they also have toxic properties against mammals, humans, and insects. Several reports describing these activities have been published by members of our research group, some of whom participate in the current submission (see doi(s): 10.1371/journal.ppat.1006946; 10.1128/AEM.01626-15; 10.111/1462-2920, 12506; 10.1094/MPMI-06-15-0127-R). However, to our knowledge, few information is available about the effect of these intermediates against insect pests. In fact, the work previously published by Rodriguez-González et al. (2018) was the first attempt to describe the effect of these intermediates on the control of Acanthoscelides obtectus, which is an important pest producing dramatic crop losses in the field and in the storage facilities. We described the toxic effect of T34-5.27 and E20-5.7 and their parental strains (T34, E20) against A. obtectus (Rodríguez-González et al., 2018). T34-5.27 and E20-5.7 are two Trichoderma harzianum derived strains which overproduce trichodiene, the first specific intermediate involved in the synthesis of the trichothecene harzianum A, whose production was described in T. arundinaceum. The latter strain was used as donor of the gene that was overexpressed in T. harzianum T34 and T. harzianum E20 strains to obtain the trichodiene overproducing transformants. Thus, in the published article Rodríguez-González et al. (2018), we studied the usefulness of trichodiene overproduction on the A. obtectus pest control.

However, in the current submitted manuscript we analyzed the effect on the A. obtectus pest control of four totally different strains: two wild type, T. arundinaceum IBT 40837 (Ta37) and T. brevicompactum IBT 40841 (Tb41), which produce the trichothecenes harzianum A and trichodermin, respectively, and the Ta37-17.139 and Ta37-23.74 mutants, derived from Ta37 by deletion of tri17 and tri23 genes, respectively. These two genes are essential for harzianum A biosynthesis (see reference 48 of the updated manuscript- also included below). As a consequence of these mutations, no one of these two mutants produce the trichothecene analog harzianum A, but both accumulate trichodermol, another intermediate in the harzianum A biosynthesis which exhibits significant in vitro phytotoxic levels (see Cardoza et al., 2015, doi: 10.1128/AEM01626-15). The difference between both mutants are the levels of trichodermol produced by them (see references 21 and 48 below). Thus, in the present manuscript we assessed the effect of the trichodermol produced by the mutated strains on the control a A. obtectus.

In conclusion, the Trichoderma strains used in these two papers (Rodríguez-González et al., 2018, and the current submission) are different, since they produce different trichothecene biosynthetic intermediates. Furthermore, we are sure that these studies are necessary to determine the putative future application of these strains or some of the trichothecene intermediates produced by them in the control of insect pests. In addition, there are other intermediates in this biosynthesis that could be analyzed in future projects. We trust that we succeeded in explaining the interest to carry out the work included in this manuscript.

  1. Proctor, R.H.; McCormick, S.P.; Kim, H.-S.; Cardoza, R.E.; Stanley, A.M.; Lindo, L.; Kelly, A.; Brown, D.W.; Lee, T.; Vaughan,

M.M.; et al. Evolution of structural diversity of trichothecenes, a family of toxins produced by plant pathogenic and entomopathogenic fungi. PLoS Pathog. 2018, 14, e 1006946. https://doi.org/10/1371/journal.ppat.1006946.

  1. Cardoza, R.E.; McCormick, S.P.; Lindo, L.; Kim, H.-S.; Olivera, E.R.; Nelson, D.R.; Proctor, R.H.; Gutierrez, S. A cytochrome P450 monooxygenase gene required for biosynthesis of the trichothecene toxin harzianum A in Trichoderma. Appl. Microbiol. Biotechnol. 2019, 103, 8087–8103. https://doi.org/10.1007/s00253-019-10047-2.

Secondly, the method section is not well described, and many details are lacking. Furthermore, the authors divided their experiments in 3 bioassays, while in fact all the three are linked and the second and third experiments are a kind of follow up of the first using material (insect and seed) resulting from that first bioassay...this raises serious doubt on the independence of the bioassays and potential biases (also in statistical analysis)!!  

Thank you for all your suggestions. More details have been added to the material and methods section, so it has been considerably improved. Like you, the rest of the reviewers thought that it is a trial linked to another one, so all the references in relation to “Experiment 1, 2, 3...” have been eliminated.

Moreover, the scientific writing is poor and in many instance does not allow understanding neither how things were done nor the rationale behind what was done. For examples, many scientific names are not italicized (see the minors below), some parts of the methodology were not well referenced (lines 156-159, see the minors below)!!  In the result section ( 3.2.2. and table 2), the authors present some findings of bioassay that was not described in the methodology!!!

Thank you for your guidance. All the scientific names have been italicized throughout the manuscript. Wrong references in the material and methods section have been changed and follow now the journal format (Page 3, lines 138 and 139). The methodology for the results in section 3.2.2 have been added to the material and methods section (Page 5, from line 204 to line 206).

Based on the above, I do not recommend publishing the Ms in its present form.

Minors:

line 40: replace "provided" with "showed"

Thank you for your suggestion. The word “provided” has been replaced to “showed” (Page 1, line 41).

line 45: replace "massive" with "large"

Thank you for your recommendation. The word “massive” has been replaced to “large” (Page 2, line 46).

line 47: put the scientific names in italic

Thank you for your suggestion. Scientific names are now in italic (Page 2, line 49).

line 54: change the word "mechanisms"

Thank you for your guidance. The word “mechanisms” has been changed to “strategies” (Page 3, line 110).

lines 50-65: As the focus of the paper is on the Trichoderma, I suggest to move this section to the end of the introduction..just before the paragraph stating the objectives of the study. Doing so, the introduction should be rearranged ( exp: lines 58-71)

Thank you for your suggestion. The paragraph has been moved to the end of the introduction, just before the paragraph stating the objectives (Page 3, from the line 106 to the line 123). The introduction has been reorganized accordingly.

lines 68-71: rephrase these sentences.

OK, thank you. Those sentences have been rephrased (Page 2, from line 70 to line 74).

line 83: correct to "synthesize"

OK, thank you. The word “synthesize” has been corrected (Page 2, line 85).

line 112: put the scientific name in italic.

OK, the words “Trichoderma” and “Trichoderma arundinaceum” have been put in italic (Page 3, line 133).

line 127: put the scientific name in italic.

OK, thank you. “A. obtectus” has been put in italic (Page 4, line 158).

Lines 136-144: I suggest to joint the 2.3 section with the 2.1 section.

Thank you for your suggestion. The 2.3 section has been joined with the 2.1 section (Page 4, from line 147 to line 155).

line 141: correct the concentration of spores.

OK, thank you. The concentration of spores has been corrected  (Page 4, line 153).

line 149: correct the concentration of spores

The concentration of spores has been corrected (Page 4, line 178).

line 150: put the scientific name in italic

Ok. The words “P. vulgaris” has been put in italic (Page 4, line 179).

lines 156-159: the methodology here described was first used by other authors (Mazzonetto and Vendramim 2003 and Fouad et al.,2012) different from the references cited here!!! The ethics rules clearly recommend to cite the original reference!!

Yes, it was wrong. Thank you for becoming aware of that. The references have been corrected and the references Mazzonetto and Vendramim 2003 and Fouad et al., 2012 have been added to the text (Page 4, lines 188 and 189).

Lines 166-169: were the petri dishes also treated with the same strain of trichoderma??

Yes, they were the Petri dishes where the treatments had been previously applied to using the Potter Tower. The explanation is on page 5, lines 199 and 200.

Line 169: delete the word "insect"

Thank you. The word “insect” has been deleted (Page 5, line 200).

Reviewer 2 Report

Review ID 1952235

 Spores of wild and transformed Trichoderma strains over P.  vulgaris beans: direct effect on Acanthoscelides obtectus attacks  and indirect effect on agronomic parameters

This is very well organized and written paper. It was my pleasure to review this manuscript.

Critical review:

1.     In my opinion the title of the manuscript is too long. I prefer rather short title. Everything what can be recognized as the main clue of the study should be explained in Abstract.

2.     Lines 50-65 contain very interesting information. However, if you present essential oils you have to mention natural defense system of plant based on volatile organic compounds!

3.     I don’t understand the aim of the study. Once you write about the main objective but in the next sentence “how to achieve the objectives”. Please clarify.

4.     I am not sure if Tables 1-3 provide enough information to build the tables. You can present them in the text.

5.     Table 5 should be presented as a Figure. You will avoid repeating "damaged" and "undamaged".

6.     Conclusions. I always prefer 2 or 3 clear sentences. Please do it.

Some other paper for consideration:

Tribolium confusum responses to blends of cereal kernels and plant volatiles

Journal of Applied Entomology 140, 558–563 (2016)

DOI: 10.1111/JEN.12284

Effect of phenolic acid content on acceptance of hazel cultivars by filbert aphid

Plant Protection Science 55(2): 116-122 (2019)

DOI: 10.17221/150/2017-PPS

Author Response

Manuscript ID: Insects - 1952235

Title: Spores of wild and transformed Trichoderma strains over P. vulgaris beans: direct effect on Acanthoscelides obtectus attacks and indirect effect on agronomic parameters

Open Review

English language and style

( ) Extensive editing of English language and style required
( ) Moderate English changes required
(x) English language and style are fine/minor spell check required
( ) I don't feel qualified to judge about the English language and style

In the last version of the attached manuscript the English has been checked by a native English speaker so the text has been improved.

Yes

Can be improved

Must be improved

Not applicable

Does the introduction provide sufficient background and include all relevant references?

( )

( )

(x)

( )

Are all the cited references relevant to the research?

( )

(x)

( )

( )

Is the research design appropriate?

(x)

( )

( )

( )

Are the methods adequately described?

(x)

( )

( )

( )

Are the results clearly presented?

( )

(x)

( )

( )

Are the conclusions supported by the results?

( )

( )

(x)

( )

Report to Reviewer 2:

Comments and Suggestions for Authors

Review ID 1952235

Spores of wild and transformed Trichoderma strains over P.  vulgaris beans: direct effect on Acanthoscelides obtectus attacks and indirect effect on agronomic parameters

This is very well organized and written paper. It was my pleasure to review this manuscript.

Dear reviewer, thank you very much for your kind words regarding the manuscript. All this helps us to continue in this research line.

Critical review:

  1. In my opinion the title of the manuscript is too long. I prefer rather short title. Everything what can be recognized as the main clue of the study should be explained in Abstract.

Thank you for your advice. The title of the manuscript has been changed to “Spores of Trichoderma strains over P. vulgaris beans: direct effect on insect attacks and indirect effect on agronomic parameters” (Page 1, from line 1 to line 4).

  1. Lines 50-65 contain very interesting information. However, if you present essential oils you have to mention natural defense system of plant based on volatile organic compounds!

Thank you for your suggestion. According to your idea, references regarding the plant's natural defense system based on volatile organic compounds has been added to the paper (Page 3, lines 117 and 118).

  1. I don’t understand the aim of the study. Once you write about the main objective but in the next sentence “how to achieve the objectives”. Please clarify.

It is true, thank you for your guidance. In order to clarify the objectives of the study, the paragraph has been simplified (Page 3, from the line 124 to the line 130).

  1. I am not sure if Tables 1-3 provide enough information to build the tables. You can present them in the text.

Thank you for your tip. Tables 1 to 3 have been removed and the information in these tables has been added to the text.

Table 1 data (page 6, from line 263 to line 278)

Table 2 data (page 10, from line 366 to line 387)

Table 3 data (page 11, from line 416 to line 426)

  1. Table 5 should be presented as a Figure. You will avoid repeating "damaged" and "undamaged".

OK, thank you for your advice Table 5 has been transformed into a figure (it is now figure 3) (from the page 17, line 547, to the page 18, line 581).

  1. Conclusions. I always prefer 2 or 3 clear sentences. Please do it.

Thank you for your guidance. The conclusions have been summarized so they were reduced now to 4 sentences (it could not be shortened more, so the different results from the study are described), and a final conclusion (from the page 20, line 704, to the page 21, line 717).

Some other paper for consideration:

Tribolium confusum responses to blends of cereal kernels and plant volatiles

Journal of Applied Entomology 140, 558–563 (2016)

DOI: 10.1111/JEN.12284

 Effect of phenolic acid content on acceptance of hazel cultivars by filbert aphid

Plant Protection Science 55(2): 116-122 (2019)

DOI: 10.17221/150/2017-PPS

Thank you for your suggestion, those papers have been added to the introduction section (Page 2, line 118).

Reviewer 3 Report

The present manuscript determined the direct and indirect effects of wild and transformed Trichoderma strains on the Acanthoscelides obtectus attacks and agronomic parameters. Some important details and information are provided in the present manuscript, but the current version need to reorganize the table and analyze the data.

(1) The English needs tightening up in various places as it is not always clear what the authors have done. The authors would benefit by giving the manuscript to someone who is very fluent in English for assistance with the grammar.

(2) The materials and methods used are standard however much more detail is required.

(3) The authors need to reorganize and analyze the data again, especially in the Tables.

(4) The writing is very rough.

Please see my comments below for specific questions/suggestions I had.

Specific comments:

Simplify the statistical analysis section.

Delete the “Experiment 1, 2 and 3” in the methods section.

Please change the “Trichoderma” to “Trichoderma” in the manuscript

Line 32: What is the species of two transformant strains of Ta37-17.139 and Ta37-23.74?

Lines 37-40: What is the mean of the Tb41 and Ta37?

Lines 40-43: Please give some data to support your description.

Lines 112-117: The description is very ambiguous. Rewrite this sentence.

Line 137: What is the components of PPG medium?

Line 139: Please change “To remove the spores” to “To collect the spores”.

Line 140: Please change “1x107 spores/ml” to “1x107 spores/ml”.

Line 152: Why use the distilled water instead of sterile water?

Line 169: Please provide the methods or the formula for calculating the mortality.

Line 153: What the mean of “the carrier in all treatments with the Trichoderma strains”?

The authors have described the methods for determine the mortality in the methods section, such as “the insect mortality of the insects in contact with the beans was recorded daily during 30 days”. However, why the authors determine the survival rate of insects exposed to beans sprayed with Trichoderma strains for 16 days in Table 1? In addition, the data presented in Table 1 are so difficult for the readers to understand which work you have done. The suggestions for me are need to use the mortality at 30 days instead of the Estimate data in Table 1, and also simplify the controls to one (all the controls are same). In addition, other tables also have the same problems.

Author Response

Manuscript ID: Insects - 1952235

Title: Spores of wild and transformed Trichoderma strains over P. vulgaris beans: direct effect on Acanthoscelides obtectus attacks and indirect effect on agronomic parameters

Open Review

English language and style

(x) Extensive editing of English language and style required
( ) Moderate English changes required
( ) English language and style are fine/minor spell check required
( ) I don't feel qualified to judge about the English language and style

In the last version of the attached manuscript, the English has been checked by a native English speaker so the text has been improved.

Yes

Can be improved

Must be improved

Not applicable

Does the introduction provide sufficient background and include all relevant references?

( )

(x)

( )

( )

Are all the cited references relevant to the research?

(x)

( )

( )

( )

Is the research design appropriate?

( )

(x)

( )

( )

Are the methods adequately described?

( )

(x)

( )

( )

Are the results clearly presented?

( )

( )

(x)

( )

Are the conclusions supported by the results?

( )

(x)

( )

( )

Report to Reviewer 3

Comments and Suggestions for Authors

The present manuscript determined the direct and indirect effects of wild and transformed Trichoderma strains on the Acanthoscelides obtectus attacks and agronomic parameters. Some important details and information are provided in the present manuscript, but the current version need to reorganize the table and analyze the data.

(1) The English needs tightening up in various places as it is not always clear what the authors have done. The authors would benefit by giving the manuscript to someone who is very fluent in English for assistance with the grammar.

(2) The materials and methods used are standard however much more detail is required.

(3) The authors need to reorganize and analyze the data again, especially in the Tables.

(4) The writing is very rough.

Please see my comments below for specific questions/suggestions I had.

Specific comments:

Simplify the statistical analysis section.

Thank you for your suggestion. The statistical analysis section has been simplified (from page 5, line 240, to page 6 line 258).

Delete the “Experiment 1, 2 and 3” in the methods section.

Thank you for your idea. The words related to “Experiment 1, 2 and 3” have been deleted in methods section (Page 5, lines 242 and 245, and page 6, line 252).

Please change the “Trichoderma” to “Trichoderma” in the manuscript

Thank you. The words “Trichoderma” are now in italic (“Trichoderma”) throughout the manuscript.

Line 32: What is the species of two transformant strains of Ta37-17.139 and Ta37-23.74?

Trichoderma arundinaceum is the Trichoderma species of the two transformed strains. An explanation has been added to the abstract section (Page 1, line 32).

Lines 37-40: What is the mean of the Tb41 and Ta37?

Ta37 means “T. arundinaceum IBT 40837 wild-type strain” and Tb41 means “T. brevicompactum IBT 40841 wild-type strain”. An explanation has been added to the abstract (Page 1, lines 31 and 34) and material and methods sections (Page 3, lines 133 and 139).

Lines 40-43: Please give some data to support your description.

OK, thank you for your guidance. More data has been added to the abstract section (Page 1, from the line 40, to the line 44).

Lines 112-117: The description is very ambiguous. Rewrite this sentence.

  1. The sentence has been shortened and rewritten to make it easier to understand (Page 3, from the line 133 to the line 139).

Line 137: What is the components of PPG medium?  

PPG solid medium contains [2% mashed potatoes (Nestlé), 2% glucose (Panreac Applichem), 2% agar (Oxoid, Ltd.)]. This description has been added to the text (Page 2, lines 147 and 148).

Line 139: Please change “To remove the spores” to “To collect the spores”.

Thank you for your suggestion. The words “To remove the spores” have been changed to “To collect the spores” (Page 4, line 150).

Line 140: Please change “1x107 spores/ml” to “1x107 spores/ml”.

Thank you for your suggestion. The words “1x107 spores/ml” have been changed to “1x107 spores/ml” (Page 4, lines 153 and 178).

Line 152: Why use the distilled water instead of sterile water?

Sterile water was used as the control treatment. Thank you for your suggestion. Now the word “distilled” has been changed to “sterile” (Page 4, line 150).

Line 169: Please provide the methods or the formula for calculating the mortality.

  1. The method (“the insects were considered dead if there was no reaction when prodded”) has been added to the Materials and Methods section (Page 5, lines 200 and 201).

Line 153: What the mean of “the carrier in all treatments with the Trichoderma strains”?

It is the support in which the spores of the fungi were diluted. This explanation has been added to the text (Page  4, lines 182 and 183)

The authors have described the methods for determine the mortality in the methods section, such as “the insect mortality of the insects in contact with the beans was recorded daily during 30 days”. However, why the authors determine the survival rate of insects exposed to beans sprayed with Trichoderma strains for 16 days in Table 1? In addition, the data presented in Table 1 are so difficult for the readers to understand which work you have done. The suggestions for me are need to use the mortality at 30 days instead of the Estimate data in Table 1, and also simplify the controls to one (all the controls are same). In addition, other tables also have the same problems.

Thank you for your suggestion. Survival has been calculated at 30 days, as described in the material and methods section. The information in Table 1 is wrong, so instead of 16 days, it should say 30 days. Also, as suggested by another reviewer, the data in table 1 has been added to the text to make it easier to understand it, and also to eliminate the amount of data that appears in the controls, much of it unnecessary. Table 1 data (page 6, from line 263 to line 278)

Round 2

Reviewer 1 Report

The authors addressed most of the raised issues.

I still would recommend a language editing by a native English speaker or by a professional.

Author Response

Manuscript ID: Insects - 1952235

Title: Spores of Trichoderma strains over P. vulgaris beans: direct effect on insect attacks and indirect effect on agronomic parameters

Open Review

English language and style

( ) Extensive editing of English language and style required
( ) Moderate English changes required
(x) English language and style are fine/minor spell check required
( ) I don't feel qualified to judge about the English language and style

In the last version of the attached manuscript the English has been checked by a native English speaker so the text has been improved.

Report to Reviewer 1 (Round 2):

Comments and Suggestions for Authors

The authors addressed most of the raised issues.

We want to thank to the reviewer for all the constructive work on this manuscript. We greatly appreciate the help.

I still would recommend a language editing by a native English speaker or by a professional.

The manuscript has been revised by a different native English professor of the Official School of Languages in Ponferrada (Spain) and, we now hope that the re-revised version of the manuscript fulfills your requirements for a well-written scientific article. (http://eoiponferrada.centros.educa.jcyl.es/sitio/).

Reviewer 2 Report

I accept corrected version of the manuscript.

Author Response

Manuscript ID: Insects - 1952235

Title: Spores of Trichoderma strains over P. vulgaris beans: direct effect on insect attacks and indirect effect on agronomic parameters

English language and style

( ) Extensive editing of English language and style required
( ) Moderate English changes required
(x) English language and style are fine/minor spell check required
( ) I don't feel qualified to judge about the English language and style

The manuscript has been revised by a different native English professor of the Official School of Languages in Ponferrada (Spain) and, we now hope that the re-revised version of the manuscript fulfills your requirements for a well-written scientific article. (http://eoiponferrada.centros.educa.jcyl.es/sitio/).

Report to Reviewer 2 (Round 2):

Comments and Suggestions for Authors

I accept corrected version of the manuscript.

Thank you very much. We want to thank to the reviewer for all the constructive work on this manuscript. We greatly appreciate the help.

Best regards,

The authors

Reviewer 3 Report

The manuscript also has some problems, although the authors have revised some of the points.

(1) The English needs tightening up in various places as it is not always clear what the authors have done. The authors would benefit by giving the manuscript to someone who is very fluent in English for assistance with the grammar.

(2) please provide the results for the effect of beans sprayed with spores of Trichoderma strains on the survival rate of insects as the tables or figures.

(3) Please provide the units for the data in Table 1.

(4) line 239: Please change the “Trichoderma” to “Trichoderma” in the revised manuscript

(5) line 177: please delete “in experiment 2”.

Author Response

Manuscript ID: Insects - 1952235

Title: Spores of Trichoderma strains over P. vulgaris beans: direct effect on insect attacks and indirect effect on agronomic parameters

Open Review

English language and style

(x) Extensive editing of English language and style required
( ) Moderate English changes required
( ) English language and style are fine/minor spell check required
( ) I don't feel qualified to judge about the English language and style

In the last version of the attached manuscript the English has been checked by a native English speaker so the text has been improved.

Yes

Can be improved

Must be improved

Not applicable

Does the introduction provide sufficient background and include all relevant references?

( )

(x)

( )

( )

Are all the cited references relevant to the research?

( )

(x)

( )

( )

Is the research design appropriate?

( )

(x)

( )

( )

Are the methods adequately described?

( )

(x)

( )

( )

Are the results clearly presented?

( )

( )

(x)

( )

Are the conclusions supported by the results?

( )

(x)

( )

( )

Report to Reviewer 3 (Round 2):

Comments and Suggestions for Authors

The manuscript also has some problems, although the authors have revised some of the points.

  • The English needs tightening up in various places as it is not always clear what the authors have done. The authors would benefit by giving the manuscript to someone who is very fluent in English for assistance with the grammar.

The manuscript has been revised by a different native English professor of the Official School of Languages in Ponferrada (Spain) and, we now hope that the re-revised version of the manuscript fulfills your requirements for a well-written scientific article. (http://eoiponferrada.centros.educa.jcyl.es/sitio/).

(2) please provide the results for the effect of beans sprayed with spores of Trichoderma strains on the survival rate of insects as the tables or figures.

Thank you for your suggestion, but in the previous revision of the manuscript, the reviewer 2 requested that survival rate data that appeared in a table be entered into the text (Page 5, from the line 219 to the line 237).

(3) Please provide the units for the data in Table 1.

Thank you for your suggestion. The units have been added to Table 1 (Page 12, line 414).

(4) line 239: Please change the “Trichoderma” to “Trichoderma” in the revised manuscript.

Thank you. The word “Trichoderma” are now in italic (“Trichoderma”) (Page 6, line 239).

(5) line 177: please delete “in experiment 2”.

Thank you for your suggestion. The words “in experiment 2” have been deleted in methods section (Page 4, line 177).